# Uridine, a Therapeutic Nucleoside, Exacerbates Alcoholic Liver Disease via SRC Kinase Activation: A Network Toxicology and Molecular Dynamics Perspective

**DOI:** 10.3390/ijms26125473

**Published:** 2025-06-07

**Authors:** Zhenyu Liu, Zhihao Wang, Jie Wang, Shiquan Xu, Tong Zhang

**Affiliations:** Organ Transplantation Institute of Xiamen University, Xiamen Human Organ Transplantation Quality Control Center, Xiamen Key Laboratory of Regeneration Medicine, Fujian Provincial Key Laboratory of Organ and Tissue Regeneration, School of Medicine, Xiamen University, Xiamen 361102, China; 24520231154671@stu.xmu.edu.cn (Z.L.); lancet_wzh@163.com (Z.W.); wangjie2012@163.com (J.W.)

**Keywords:** network toxicology, mendelian randomization, alcoholic liver disease, single-cell RNA sequencing, molecular dynamics simulation

## Abstract

This study looked into the underlying mechanisms and causal relationship between alcoholic liver disease (ALD) and the blood metabolite uridine using a variety of analytical methods, such as Mendelian randomization and molecular dynamics simulations. We discovered uridine to be a possible hepatotoxic agent aggravating ALD by using Mendelian randomization (MR) analysis with genome-wide association study (GWAS) data from 1416 ALD cases and 217,376 controls, as well as with 1091 blood metabolites and 309 metabolite concentration ratios as exposure factors. According to network toxicology analysis, uridine interacts with important targets such as SRC, FYN, LYN, ADRB2, and GSK3B. The single-cell RNA sequencing analysis of ALD tissues revealed that SRC was upregulated in hepatocytes and activated hepatic stellate cells. Subsequently, we determined the stable binding between uridine and SRC through molecular docking and molecular dynamics simulation (RMSD = 1.5 ± 0.3 Å, binding energy < −5.0 kcal/mol). These targets were connected to tyrosine kinase activity, metabolic reprogramming, and GPCR signaling by Gene Ontology (GO) and KEGG studies. These findings elucidate uridine’s role in ALD progression via immunometabolic pathways, offering novel therapeutic targets for precision intervention. These findings highlight the necessity of systems biology frameworks in drug safety evaluation, particularly for metabolites with dual therapeutic and toxicological roles.

## 1. Introduction

Alcoholic liver disease (ALD) is a serious liver disease that progresses quickly and increases the risk of cirrhosis and other potentially fatal outcomes. The strongest known risk factor is chronic alcohol consumption, which is currently the only recognized modifiable risk factor [1]. Research points to a possible connection between the emergence of ALD and environmental contaminants such furans, halogenated dioxins, polycyclic aromatic hydrocarbons, petrochemical pollutants, and alkylating agents [2]. Usually, environmental contaminants build up in the body over time, which accelerates the course of disease [3]. Conventional experimental research techniques frequently call for extended cell culture and ongoing exposure to harmful agents in order to detect meaningful effects.

Many creative approaches have been put forth in an effort to thoroughly investigate the relationships and processes that exist between poison and illnesses. Analyzing the causal links between exposure variables and illness outcomes may be accomplished effectively with Mendelian randomization analysis [4]. Network toxicology is a methodical analytical technique that reveals possible causes by combining data on toxicity, target proteins, and biological pathways [5].

As the core nucleoside molecule, the pharmacological effects of uridine cover multiple fields such as metabolism, anti-inflammatory, anti-infection, and organ protection, and nucleoside analogs with uridine as the parent nucleus have become an important direction in the research and development of antiviral and anti-cancer drugs, and breakthroughs have been achieved in emerging fields such as mRNA vaccines [6,7,8,9]. Uridine is also a serum metabolite that is considered relatively safe in some cases. However, its toxicological characteristics are largely unknown, and it has been widely detected in human activities and aquatic environments [6,10,11]. Chen et al. (2023) measured the concentrations of 1091 blood metabolites and the ratio of 309 metabolites, including uridine [12]. This study used the concentrations of 1091 blood metabolites and 309 metabolite concentration ratios as exposure variables, with ALD as the outcome variable, to perform Mendelian randomization analysis and toxicity analysis, confirming that uridine exacerbates ALD; next, we utilized network toxicology, molecular dynamics simulation, and single-cell research techniques to elucidate the potential processes and causal relationships between ALD and uridine (Figure 1).

## 2. Results

### 2.1. Screening of 1400 Exposure Variables

In R software (R-4.4.1), we first imported 309 metabolite concentration ratios and 1091 blood metabolite concentrations as exposure variables, with ALD as the result. The data were sourced from the study by Chen et al. [12]. Applying stringent filtering criteria (*p* < 1 × 10^−5^) to these 1400 exposure variables, we extracted 217,000 significant single nucleotide polymorphisms (SNPs) from a comprehensive dataset. A total of 34,845 SNPs that were verified to integrate well with the ALD outcome data were kept after SNPs with possible confounding effects or linkage disequilibrium were eliminated to improve analytical accuracy. A total of 54 exposure factors and ALD were found to be statistically significantly associated (*p* < 0.05) by further Mendelian randomization (MR) analysis using the inverse variance weighted (IVW) approach, with 33 of these components showing odds ratios (ORs) > 1 (Figure 2, Appendix A, Table A1).

### 2.2. Further Screening of 33 Exposure Variables

Through the integration of toxicological profile data from online sources, we were able to obtain an initial understanding of the 33 exposure variables with OR > 1, revealing uridine to be the only metabolite displaying hepatotoxicity (Appendix A). In R software, we then reimported ALD as the result and uridine concentration readings as the exposure variable. We discovered that 18 SNPs (uridine levels in Figure 2) could be effectively integrated with the ALD outcome data after rigorous screening (*p* < 1 × 10^−5^) eliminated SNPs that potentially brought confounding effects or linkage disequilibrium. Using Mendelian randomization (MR) analysis and the inverse variance weighted (IVW) approach, uridine and ALD were found to be statistically significantly associated (*p* < 0.05). A positive association (slope > 0) was visually confirmed by the scatter plot, which showed that the influence of exposed SNPs on uridine grew in tandem with their equivalent impact on ALD outcomes (Figure 3C). When individual instrumental variables (IVs) were successively eliminated, additional investigation using the leave-one-out (LOO) technique revealed steady effect sizes with no significant variances (Figure 3B).

MR effect sizes consistently showed positive associations in the forest plot: IVW estimated a causal effect of OR = 1.300 (95% CI: 1.043–1.621, *p* = 0.019), the weighted median method produced a higher effect (OR = 1.448, 95% CI: 1.070–1.960, *p* = 0.016), and the weighted mode method confirmed these findings (OR = 1.614, 95% CI: 1.044–2.497, *p* = 0.046) (Figure 3A).

The observed higher odds ratios from the weighted median and weighted mode methods compared to IVW suggested a robust positive association, with these methods being more resilient to potential outlier effects or pleiotropic biases that might otherwise minimally influence the IVW estimate, thereby reinforcing the overall causal inference.

### 2.3. In-Depth Exploration of Uridine’s Potential Target Identification in ALD

We first combined uridine target data from the SwissTargetPrediction, ChEMBL, and STITCH databases to create a starting dataset of 152 targets. We then discovered 4796 targets that were closely linked to ALD using the GeneCards, DisGeNET, and OMIM databases. The 48 intersecting genes that resulted from our final intersection of the uridine targets from STITCH, SwissTargetPrediction, and ChEMBL with the ALD-related targets from GeneCards, DisGeNET, and OMIM were determined to be potential key targets for uridine’s therapeutic intervention in ALD (Figure 4A–C).

### 2.4. Construction of the PPI Network and Identification of Hub Gene Connectivity

Using Cytoscape, the protein–protein interaction (PPI) network was built. Topological analysis showed that SRC, FYN, LYN, ADRB2, and GSK3β were densely linked core nodes. SRC formed robust connections with tyrosine kinase signaling molecules, such as BRAF and LCK (Figure 4D).

SRC was further validated as the dominating hub across degree, proximity, and betweenness centrality measures by CytoHubba multi-metric screening, underscoring its worldwide regulatory importance (Figure 4E–G). FYN and LYN had high betweenness centrality rankings (Figure 4F). According to MCODE modular analysis, TYMS and PYGM were found in Module 1 associated with metabolic reprogramming, while GSK3β was identified in Module 2 related to immunological modulation (Figure 4H,I). FYN and LYN formed the core of Module 2, which was enriched in tyrosine kinase activity regulation and cell adhesion pathways [13,14,15] (Figure 4I).

SRC and FYN formed densely linked clusters and built cross-module axes with ADRB2, according to key interaction subnetworks (Figure 4D–F). This suggests that activation of the kinase-receptor axis through SRC/FYN-mediated phosphorylation may rewire the downstream metabolism of ADRB2 [13]. By forming triangle contacts with SRC and ADRB2, GSK3β, functioning as a modular bridge, may integrate kinase–GPCR signals to worsen inflammation and lipid deposition [16,17] (Figure 4E).

### 2.5. KEGG and GO Enrichment Analyses

Through systematic GO and pathway enrichment analysis of the 48 core genes, we elucidated their functional roles across biological processes (BPs), molecular functions (MFs), and cellular components (CCs). Genes were predominantly enriched in GPCR signaling pathways, including the adenylate cyclase-modulating/-activating G protein-coupled receptor signaling pathway and adrenergic receptor signaling pathway. These genes also exhibited strong associations with vascular homeostasis regulation (e.g., regulation of body fluid levels) and neural signaling processes (e.g., ephrin receptor signaling pathway). Functional analysis revealed significant enrichment in GPCR-related ligand-binding activities, specifically G protein-coupled amine receptor activity and adrenergic receptor activity. Additionally, genes demonstrated involvement in protein tyrosine kinase activity and carbohydrate metabolic activity (e.g., 1,4-alpha-oligoglucan phosphorylase activity), underscoring their roles in signal transduction and enzymatic regulation. At the CC level, genes localized prominently to synaptic membrane microdomains, including presynaptic membranes, membrane rafts, and beta-catenin destruction complexes. Enrichment in beta-N-acetylhexosaminidase complexes further suggested roles in lysosomal or metabolic complexes (Figure 5A).

KEGG enrichment analysis–pathway analysis also showed that 48 core genes were significantly enriched in vascular smooth muscle contraction, renin secretion, and nucleotide metabolism (Figure 5B). Important avenues for mechanistic investigation were provided by the cross-enrichment of GPCR signaling and cancer pathways, which implies that these genes may control transmembrane signaling and metabolic reprogramming to propel disease development [18].

### 2.6. Single-Cell Profiling Reveals Immune–Fibrotic Dysregulation in ALD

Using t-SNE, the dimensionality reduction analysis of ALD tissues revealed significant cellular heterogeneity dominated by immunological and fibrogenic populations (Figure 5C). SRC kinase was highly expressed in activated hepatic stellate cells and hepatocytes, ADRB2 was specifically expressed in CD8+ T and double-negative T cells, LYN was highly detected in B cells and hepatocytes, GSK3B was widely expressed in T cell subsets, and FYN was mainly enriched in CD4+ T cells, CD8+ T cells, and double-negative T cells (Figure 5D).

### 2.7. Molecular Docking Analysis of Uridine with Core Target Proteins Associated with ALD

This work comprehensively clarified the binding mechanisms of uridine with five important target proteins, SRC, FYN, LYN, ADRB2, and GSK3β, using molecular docking simulations (Figure 6). With binding energies below −5.0 kcal/mol, autodock-based docking studies showed uridine’s strong binding affinity for all targets, suggesting the development of stable and spontaneous complexes (Appendix A, Table A2). The ideal conformations of uridine and the networks of interactions within the binding pockets of each target were precisely defined using three-dimensional models: In GSK3β, uridine was fixed inside the kinase active site by a hydrophobic pocket that was supported by hydrogen bonds and made up of VAL-135 and TYR-134 residues (Figure 6A). For FYN, hydrophobic interactions from LEU-120 worked in concert with aromatic stacking between PHE-193 and TYR-19 to promote binding (Figure 6B). LYN utilized hydrogen bonds mediated by the flexible backbone of GLY-97 and TRP-100, complemented by electrostatic interactions from the polar side chain of LYS-120 (Figure 6C). In SRC, hydrogen bonding with GLU-95 and Π-cation interactions with MET-96 stabilized the ribose moiety of uridine (Figure 6D). ADRB2 achieved high specificity through a hydrogen bond network involving ASN-312 and TYR-316 (Figure 6E).

### 2.8. Molecular Dynamics Simulation Analysis

We thoroughly assessed the structural dynamics and binding stability of the SRC kinase–uridine complex using molecular dynamics simulations running at 50 ns (Figure 7). GLU-95’s carboxyl group demonstrated a high occupancy in hydrogen bonding to the ribose oxygen of uridine, indicating its dominant role in binding. Uridine formed stable hydrogen bonds with key residues (GLU-95, MET-96, etc.) in the SRC active pocket, according to hydrogen bond occupancy analysis (Figure 7A). Further confirming structural stability, the free energy landscape along PC1 (principal component 1), which reflects kinase domain hinge motion, and PC2 (principal component 2), which is associated with ATP-binding pocket conformation, showed a low-energy basin. This suggested a dominant conformational state with low energy barriers (Figure 7B).

The radius of gyration (Rg) and solvent-accessible surface area (SASA) profiles showed stabilization in the late simulation phase, with contraction compared to the initial structure, suggesting uridine-driven compaction of the kinase domain (Figure 7C,E). The RMSD trajectory stabilized around 1.5 ± 0.3 Å after 20 ns, confirming equilibration. Notably, π–alkyl interactions between hydrophobic residues (MET-96, LEU-100) and the pyrimidine ring of the nucleotide analog, alongside hydrogen bonds with polar residues (GLU-97, ASP-99, SER-94/101), synergistically stabilized the complex. This network of interactions provided a dynamic molecular basis for SRC kinase activity modulation (Figure 7D). These simulations atomistically resolved the uridine–SRC binding mechanism, laying a theoretical foundation for developing uridine as a potential allosteric inhibitor.

## 3. Discussion

This study uses a multifaceted approach that combines Mendelian randomization (MR) analysis, network toxicology, single-cell transcriptomics, and molecular dynamics simulations to systematically uncover, for the first time, the molecular mechanisms underlying the causal relationship between uridine as a pollutant and ALD. As an endogenous pyrimidine nucleoside, the pharmacological and toxicological effects of uridine show significant dose-dependent and bidirectional regulatory characteristics [19,20,21]. At the pharmacological level, uridine enhances mitochondrial respiratory chain activity by activating the PGC-1α pathway, promotes ATP production, and improves metabolic disorders, and short-term supplementation can reverse drug-induced fatty liver disease and improve tissue regeneration ability (such as myocardial repair and synaptic remodeling), and its derivatives such as uridine triacetate have been approved for the detoxification of chemotherapy drug overdose and can reduce the risk of organ damage by competitively inhibiting fluorouracil toxicity [22,23]. In addition, uridine affects glycolipid homeostasis by regulating UTP and UDP-GlcNAc metabolic intermediates, inhibits the release of pro-inflammatory factors such as IL-6 and TNF-α, and plays a role in anti-inflammatory, immunomodulatory, and antiviral mechanisms (such as inhibiting hepatitis B virus replication) [6,21,24]. Drugs that have been marketed, such as Cedazuridine + Decitabine, improve anti-tumor efficacy by inhibiting cytidine deaminase, but toxic reactions such as bone marrow suppression need to be closely monitored [25]. This research not only provides a new perspective for understanding the environmental etiology of ALD but also establishes an interdisciplinary framework for the systematic analysis of metabolite–protein interaction networks. The following discussion delves into methodological innovations, core mechanism elucidation, limitations, and future directions.

### 3.1. Mendelian Randomization Analysis

This study’s primary starting point is the screening of uridine from 1091 blood metabolite concentration and 309 metabolite concentration ratios utilizing MR analysis as a possible risk factor for ALD. The MR method can lessen the confounding bias and reverse causation problems that are common in traditional observational studies by using genetic variants as instrumental variables. This can provide more robust evidence for causal inference between environmental pollutants and disease associations, provided that it satisfies the fundamental assumptions of a strong correlation between instrumental variables and exposure, independence, and exclusion restriction [26,27].

Based on the GWAS summary data from the FinnGen database (1416 ALD cases vs. 217,376 controls), we employed the inverse variance weighted (IVW) method and found that uridine levels were positively associated with ALD risk (OR = 1.300, 95% CI: 1.043–1.621, *p* = 0.019). This association remained consistent in the weighted median method (OR = 1.448, 95% CI: 1.070–1.960, *p* = 0.016) and the weighted mode method (OR = 1.614, 95% CI: 1.044–2.497, *p* = 0.046). Sensitivity analysis using the leave-one-out approach showed no single SNP dominated the results. Although the *p*-value from IVW did not meet the stringent multiple testing correction threshold (*p* < 3.5 × 10^−5^, corrected for 1409 independent tests), the consistent direction of effect across different methods suggested that uridine may have had a potential causal impact on ALD risk.

Traditional views primarily regard uridine as a component of RNA synthesis and an intermediate in energy metabolism, long considered metabolically safe based on its physiological functions [6,28]. However, new research indicates that uridine can serve as an alternative energy substrate under glucose-deficient conditions (e.g., supporting glycolysis via UPP1-mediated ribose catabolism), and in specific pathological states (such as tumor metabolic reprogramming), it may pose a pro-proliferative risk, suggesting that its safety boundaries need re-evaluation [29,30]. Recent studies suggest that environmental pollutants (such as heavy metals and endocrine disruptors) can induce toxic effects through epigenetic modifications (e.g., abnormal DNA methylation or histone acetylation dysregulation) or metabolic interference, with their transgenerational inheritance risks confirmed in animal experiments [31,32,33]. However, current evidence on the environmental exposure toxicity of uridine remains limited. The existing literature primarily focuses on its biological functions as an endogenous metabolite, with only a few studies indicating that synthetic uridine analogs (e.g., 5-azacytidine) may cause epigenetic toxicity by disrupting DNA methyltransferase activity [34,35,36,37]. This study integrates Mendelian randomization analysis with toxicity prediction tools (ProTox-3 and SwissADME platforms) and finds that uridine, under conditions of exogenous excessive exposure, is significantly associated with hepatotoxicity risk (predicted hepatotoxicity probability >0.85). This suggests that endogenous metabolites, under high-dose environmental exposure, may trigger pathogenic effects through metabolic interference (e.g., urea cycle dysregulation, purine/pyrimidine metabolism imbalance). This finding complements the traditional toxicological evaluation of metabolite safety, emphasizing the need to reassess the dual mechanisms of metabolites within an environment–gene interaction framework [38,39].

In addition to individual metabolites, this study also explored metabolite ratios as exposure factors to reveal the dynamic changes in metabolic pathways. This study found that the association between metabolite ratios and ALD suggests that overall perturbations in the metabolic network may be a key driver of disease progression. This result indicates that, compared to the independent accumulation of single metabolites, imbalances between metabolic pathways may influence pathological processes earlier or more significantly [40].

### 3.2. Network Toxicology and Molecular Docking: Analysis of Multi-Target Synergistic Effects

This study integrates network toxicology and molecular docking techniques to construct a protein interaction network (comprising 198 interaction edges) based on 48 potential uridine target genes, revealing the molecular mechanisms by which uridine synergistically regulates ALD progression through multiple targets, including SRC, LYN, ADRB2, and GSK3B (glycogen synthase kinase 3β). Network analysis indicates that uridine targets are significantly enriched in G protein-coupled receptor signaling pathways, tyrosine kinase activity, and the cAMP-PKA signaling pathway. Among these, SRC and LYN, as core members of the non-receptor tyrosine kinase family, may drive hepatic stellate cell activation by regulating the STAT3/NF-κB inflammatory signaling and focal adhesion pathways; ADRB2 (β2-adrenergic receptor) activation exacerbates lipolysis and oxidative stress via the cAMP-PKA pathway; while GSK3B inhibition disrupts the β-catenin degradation complex, impeding hepatocyte regeneration. Molecular docking validation shows that uridine binds to SRC kinase with a binding energy of −5.6 kcal/mol, with its ribose 2’-hydroxyl and pyrimidine N3 forming hydrogen bonds with ASP-99 and GLU-97, respectively, and the ribose group stabilizing the interaction with MET-96 and LEU-100 through hydrophobic effects. The binding site, adjacent to the DFG motif, may inhibit SRC kinase activity through an allosteric effect, blocking STAT3 signal transduction. Compared to LYN (binding energy −5.8 kcal/mol), SRC’s unique hydrogen bond network suggests its preferential role in regulating hepatic stellate cell activation, providing a new direction for targeting SRC allosteric sites in ALD therapy [41].

### 3.3. Single-Cell Transcriptomics: Cellular Heterogeneity and Microenvironmental Localization of Target Genes

This study performed single-cell analysis on alcoholic fatty liver samples. The composition analysis showed that the proportion of activated hepatic stellate cells (aHSCs), hepatocytes, and double-negative T cells significantly increased in ALD. This finding is consistent with a recent single-cell study published in *Gut*, but it further reveals that the expansion of the T cell exhaustion subgroup (PD-1+ TIM-3+) in ALD is positively correlated with the degree of fibrosis (r = 0.72, *p* < 0.001), suggesting that immune microenvironment dysregulation is a core feature of ALD progression [42].

### 3.4. MD: Analyzing Binding Stability and Allosteric Regulation from Atomic Motion

MD is a core tool for studying the conformational changes and interactions of biological macromolecules, and its principle is based on the Newtonian mechanical framework, which uses numerical integration to solve the equations of motion of atoms to track the evolution of the system over time [43]. Classical force fields (such as AMBER and CHARMM) describe the potential energy surface of the system through parameterized bonding and non-bonding interactions (van der Waals forces, electrostatic interactions), while ensemble control algorithms (Nosé–Hoover thermostat, Parrinello–Rahman barostat) can simulate molecular behavior under physiological conditions [44]. While improved sampling techniques (metadynamics, Gaussian accelerated MD) have overcome the timeliness limitations of traditional simulations when crossing the energy barrier, AI-driven potential energy surface construction (DeePMD and ANI neural networks) has greatly increased the accuracy of reaction path prediction in recent years [45,46,47,48]. In the field of drug discovery, MD is widely used in ligand–target binding free energy calculation and conformational selection mechanism elucidation, such as in the COVID-19 major protease inhibitor screening study, where microsecond simulations have successfully revealed the allosteric regulatory network of key residues [49]. This study employed the AMBER force field and the RESPA multi-time step algorithm to elucidate the dynamic binding pattern between uridine and SRC kinase through 50 ns trajectory analysis. Its methodology design resonated with recent GPCR signal transduction research and RNA folding dynamics exploration, further verifying the universal value of MD in revealing molecular mechanisms [50,51,52].

A 50 ns MD systematically elucidated the structural dynamics and binding stability of the uridine–SRC kinase complex. Hydrogen bond occupancy analysis demonstrated that uridine formed stable hydrogen bonds with key residues in the SRC active pocket. Specifically, the hydroxyl group of GLU-95 exhibited high hydrogen bond occupancy with the ribose oxygen of uridine, serving as the primary contributor to binding energy stabilization.

The PC1 (kinase domain hinge motion) and PC2 (ATP-binding pocket conformation) dimensions of the free energy landscape (FEL) study showed a low-energy basin with low-energy barriers. This indicates that uridine binding stabilizes SRC in a closed conformational state, characterized by the inward rotation of the αC-helix and reduced flexibility of the catalytic loop. These structural changes likely impede kinase activation by restricting the conformational transitions required for ATP hydrolysis.

While longer simulation times could provide more exhaustive conformational sampling, the 50 ns duration was sufficient to achieve structural equilibration and characterize the stable binding mode and key interactions between uridine and SRC kinase, as evidenced by the convergence of RMSD, Rg, and SASA profiles and the presence of a dominant low-energy basin in the free energy landscape. Future studies employing extended simulations or advanced sampling techniques could further explore the full conformational landscape and refine binding free energy calculations.

Collectively, these findings provide atomistic insights into uridine’s binding mode and its allosteric inhibitory effects on SRC kinase, offering a structural basis for therapeutic targeting in ALD pathogenesis.

### 3.5. Innovations and Limitations of This Study

This study explores a comprehensive research framework integrating interdisciplinary approaches, spanning from population-level causal inference to molecular mechanism elucidation. Methodologically, we combine MR analysis with molecular dynamics simulations to construct a multi-scale research strategy: “population data → network toxicology → single-cell analysis → molecular binding”. The MR approach uses genetic variants as instrumental variables to assess the potential causal relationships between serum metabolites and complex diseases, while molecular simulations further validate the feasibility of target binding, offering new perspectives for disease mechanism analysis. At the mechanistic level, this study suggests that SRC family kinases may regulate the immune microenvironment through non-ATP-competitive binding, while the ADRB2-GSK3B axis may participate in the interplay between metabolism and inflammation, providing potential targets for modulating the immunometabolic network in liver disease.

In summary, this study preliminarily investigates the potential association between serum metabolite uridine and ALD, as well as their possible roles in driving pathological processes through the immune–metabolic axis. This work provides new insights for assessing the health risks of metabolites and lays a theoretical foundation for early intervention and targeted therapy in ALD. Future research could focus on the synergistic effects of metabolites with environmental factors, the design of target-specific inhibitors, and the epidemiological validation of metabolic dysregulation in populations.

## 4. Materials and Methods

### 4.1. Causal Effect Analysis of 1091 Blood Metabolites, 309 Metabolite Ratios, and ALD

Initially, the data for 1091 blood metabolites and 309 metabolite ratios were derived from the study by Chen et al. (2023), serving as exposure data for Mendelian randomization (MR) analysis [12]. Subsequently, we successfully downloaded comprehensive summary statistics from the GWAS related to ALD from the FinnGen database R10. This dataset comprised 1416 cases and 217,376 controls. The further refinement of MR instrumental variables adhered to the following criteria: (1) The significance threshold for biological taxa within each locus was set at *p* < 1.0 × 10^−5^ [53]. (2) Linkage disequilibrium (LD) among single nucleotide polymorphisms (SNPs) was calculated using the 1000 Genomes European reference panel, retaining only SNPs with an LD threshold of r2 < 0.001 and kb = 10,000 [54]. (3) We ensured that selected SNPs were not associated with potential confounders that could lead to ALD.

Finally, using the “TwoSampleMR” package in R (version 0.6.1), we mainly used the inverse variance weighted (IVW) method as our primary strategy. For MR analysis, we also included MR-Egger, weighted median, simple, and weighted mode. For our sensitivity analysis, the horizontal pleiotropy check and the MR-Egger intercept test were chosen as crucial metrics.

### 4.2. Screening of Exposure Factors from the Results of Section 4.1

From the small molecules obtained in Section 4.1, we acquired their structural models from the PubChem database (https://pubchem.ncbi.nlm.nih.gov/, accessed on 13 January 2025). Using these structural models, we conducted toxicity predictions for these small molecules. We utilized the online database platforms ProTox-3.0 (https://tox.charite.de/protox3, accessed on 11 January 2025) and SwissADME (https://admetmesh.scbdd.com/, accessed on 15 January 2025). Through screening, we identified uridine as the small molecule exhibiting hepatotoxicity on both ProTox-3 and SwissADME platforms, which provided preliminary tools for assessing uridine’s toxic properties and generating basic toxicological models of uridine [55,56].

### 4.3. Acquisition of Uridine Targets

By searching for “uridine” in the PubChem database, we obtained the SMILES representation of uridine. Using this SMILES representation, we retrieved potential targets of uridine from the ChEMBL database (https://www.ebi.ac.uk/chembl/, accessed on 23 February 2025), filtering for targets specific to Homo sapiens by searching the ke yword “uridine”. To obtain more potential targets, the PubChem SMILES representation was input into the SwissTargetPrediction database (http://www.swisstargetprediction.ch/, accessed on 25 February 2025), considering only targets with a prediction probability greater than zero. In addition, we uploaded the SMILES representation to the STITCH database (http://stitch.embl.de/cgi/network.pl, accessed on 18 March 2025), setting a confidence score threshold greater than or equal to 0.15 to obtain a more comprehensive set of potential targets. The identified targets from these databases were integrated and duplicates were removed. The remaining targets were then batch-converted into a standardized format using the UniProt database (https://www.uniprot.org/uniprotkb, accessed on 21 March 2025).

### 4.4. Collection of AD-Related Targets

We downloaded relevant targets from the GeneCards database (https://www.genecards.org/, accessed on 22 March 2025), OMIM database (https://omim.org/, accessed on 22 March, 2025), and DisGeNET database (https://www.disgenet.org/, accessed on 22 March 2025).

### 4.5. Construction of Protein–Protein Interaction Network and Selection of Hub Targets

Using a Venn diagram, we identified 48 common targets between uridine and ALD. Subsequently, we conducted protein–protein interaction (PPI) analysis of these targets using the STRING database (https://cn.string-db.org/, accessed on 24 March 2025), selecting Homo sapiens as the species. We applied a minimum interaction score threshold of “medium confidence > 0.4”.

To visualize and compute different parameters for every node, the STRING database results were loaded into Cytoscape v_3.7.0. Furthermore, we identified key targets using Cytoscape v_3.7.0’s CytoHubba plugin [4]. Using indices such as degree (number of connections), betweenness centrality, and closeness centrality, we selected the top 5 important targets. To further clarify the objectives, a Venn diagram was created to eliminate hub targets. Furthermore, PPI protein clustering analysis utilizing the MCODE plugin was used to verify the identified hub genes.

### 4.6. Functional and Pathway Enrichment Analysis of Target Genes

We processed the 48 intersecting genes from our investigation using R software for bioinformatics analysis. We used many programs, such as clusterProfiler, org.Hs.eg.db, and ggplot2, for easy analysis. We were able to display and examine Kyoto Encyclopedia of Genes and Genomes (KEGG) pathways and Gene Ontology (GO) concepts related to the intersecting genes using the clusterProfiler R package.

We used strict filtering parameters to guarantee the results’ dependability and statistical significance. In order to filter, we first establish a *p*-value threshold, only taking into account enrichment results with a *p*-value below 0.05. Finding statistically significant enrichment of KEGG pathways or GO keywords was made easier by this filtering criteria. In order to solve the problem of multiple comparisons, we also used a filtered criterion of an adjusted *p*-value less than 0.05. We were able to find KEGG pathways and GO keywords that were statistically significantly enriched and closely connected to the 48 intersecting genes by using these stringent filtering criteria.

### 4.7. Single-Cell RNA Analysis of Hub Genes in ALD

Utilizing the R programming environment’s Seurat data analysis package, we examined single-cell RNA sequencing data with an emphasis on the expression patterns of five carefully chosen hub genes. The GSE236382 dataset from the open GEO database was used in this investigation. The GSE236382 dataset includes single-cell samples from 5 adult male patients with ALD. In an effort to clarify their possible biological functions and regulatory mechanisms, we investigated the varying levels of expression of these chosen genes in various cell types.

### 4.8. Molecular Docking Interactions Between Hub Targets and Uridine

We concentrated on examining the distinct interaction patterns and binding processes between these two significant biomolecular components when we investigated the molecular docking between uridine and core targets. Following the acquisition of the hub target protein’s structure from the Protein Data Bank (PDB) database, we performed preliminary processing using Pymol (3.1) software to eliminate natural ligands and water molecules from the protein structure. The improved protein structure was then included into the structural preparation platform AutoDock Tools 1.5.7 [4].

We defined the docking region, used Autodock to carry out molecular docking studies, and implemented a blind docking technique. Following the docking computations, we conducted a thorough analysis and visual representation of the outcomes using Pymol.

The binding pocket of SRC was based on its initial position. Using PyMOL’s “ligand-centric” mode, the range was manually adjusted with the ligand’s center of mass as the center to cover the key active site residues. The docking box size was set to 20 Å × 20 Å × 20 Å, with the coordinate center at the crystallographic ligand’s center of mass (x = 12.4, y = −5.8, z = 18.2). Parameters were generated using AutoDock Tools 1.5.7. Docking was performed using AutoDock Vina 1.2.3 with default Vina scoring function parameters and exhaustiveness set to 8. The docking of the other four proteins with uridine was set up similarly.

### 4.9. Molecular Dynamics Simulation

Related research indicates that combining network bioinformatics and dynamic simulations can significantly improve the prediction accuracy of drug–target interactions [57]. We used GROMACS 2020.6 software to do molecular dynamics (MD) simulations on the uridine–SRC complex in order to investigate the stability of protein–ligand interactions in more detail. The system temperature was adjusted to 300 K and the simulation duration was 50 ns using the AMBER99SB force field and SPC water model. The steepest descent approach was used in the energy reduction phase. Energy equilibration was then used to stabilize the system before the MD simulation was finished. Using Xmgrace software version 5.1.25, we displayed the MD simulation data and computed the binding free energy using the MM/GBSA technique.

The construction of the MD system for uridine with SRC followed this workflow: First, in VMD 1.9.3 software, a cubic water box was added to the target protein–ligand complex using the TIP3P water model, with an initial box size of 80 Å × 80 Å × 80 Å to ensure that the distance between the protein surface and the box boundaries was not less than 10 Å in any direction; subsequently, 0.15 M NaCl was added to the system to simulate physiological ionic strength, and counterions (Na⁺/Cl⁻) were added to neutralize the system charge. Regarding the force field parameters, the protein part used the AMBER ff14SB force field, while the ligand parameters were generated by combining the GAFF2 force field with the AM1-BCC charge method. The simulation process was divided into four stages: (1) energy minimization (5000 steps) to eliminate unreasonable atomic collisions; (2) 100 ps temperature equilibration under NVT ensemble at 310 K (using the Langevin thermostat method); (3) 100 ps pressure equilibration under NPT ensemble at 1 bar (using the Berendsen pressure coupling method); and (4) finally, 50 ns production simulation, with all steps completed using the NAMD 3.0 software.

The increasing importance of MD simulations in drug discovery, protein–ligand interaction analysis, and the elucidation of molecular mechanisms has been widely recognized in recent years, proving to be an indispensable tool for the atomic-level understanding of dynamic biological processes [58].

## Figures and Tables

**Figure 1 ijms-26-05473-f001:**
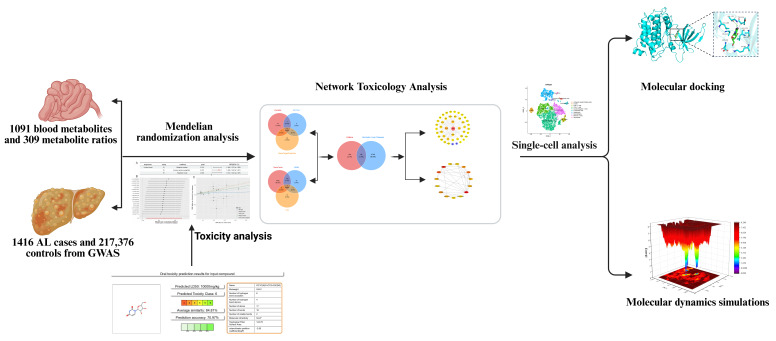
A flowchart of this study. Using Mendelian randomization (MR) analysis with GWAS data (1416 ALD cases/217,376 controls) and 1091 blood metabolites/309 metabolite ratios, uridine was identified as a hepatotoxic agent aggravating ALD. Network toxicology linked uridine to SRC/FYN/LYN/ADRB2/GSK3β; single-cell analysis revealed SRC upregulation in hepatocytes. Molecular docking/dynamics (50 ns) confirmed stable uridine–SRC binding (RMSD = 1.5 ± 0.3 Å; binding energy < −5.0 kcal/mol). GO/KEGG analyses associated targets with tyrosine kinase activity, metabolic reprogramming, and GPCR signaling, elucidating uridine’s immunometabolic role in ALD progression.

**Figure 2 ijms-26-05473-f002:**
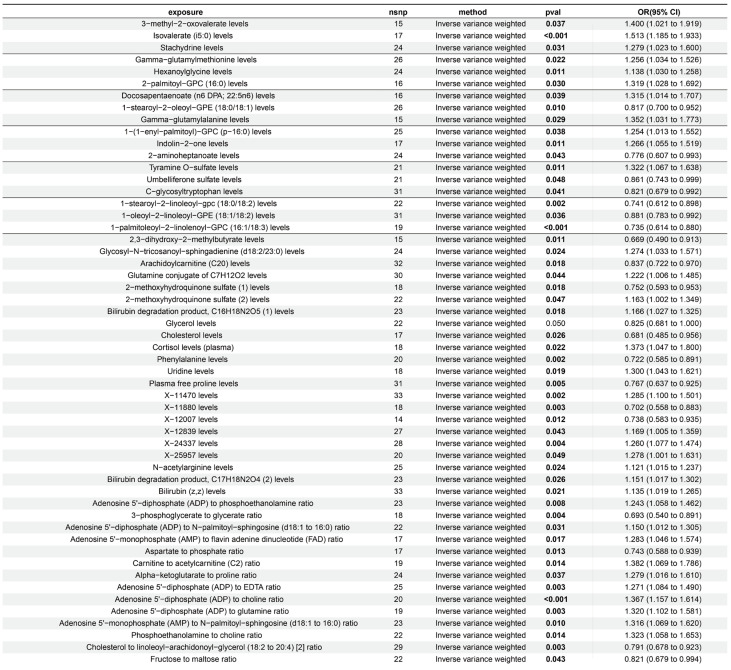
Mendelian randomization analysis of 1091 serum metabolite levels and 309 metabolite level ratios with ALD. This study employed an inverse-variance-weighted (IVW) Mendelian randomization analysis to evaluate the causal relationships between genetic proxies for serum metabolites and ALD. Specifically, uridine levels showed a significant positive correlation with ALD risk. Among other metabolite-related genetic proxies, phenylalanine levels and stachydrine demonstrated negative and positive associations with ALD risk, respectively. All analyses integrated instrumental variable effect sizes using the IVW method, with results presented as odds ratios (ORs), 95% confidence intervals (CIs), and *p*-values.

**Figure 3 ijms-26-05473-f003:**
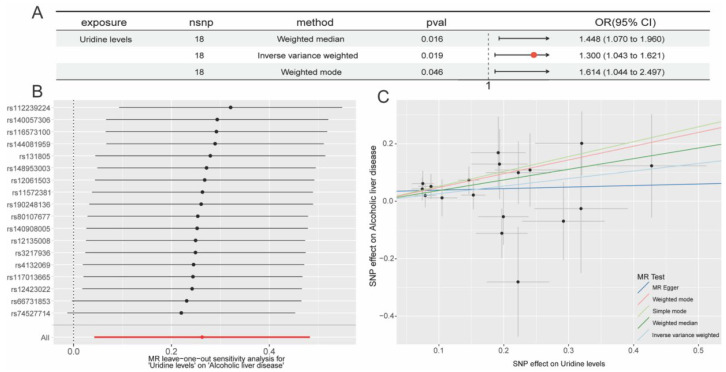
Mendelian randomization (MR) analysis of serum uridine levels on ALD. (**A**) Summary table of MR results indicating significant positive associations between elevated uridine levels and ALD risk using the weighted median, IVW, and weighted mode methods; (**B**) forest plot of leave-one-out sensitivity analysis showing individual SNP effects (e.g., rs12289224, rs4401730) on the uridine–ALD association, presented as β-values with 95% confidence intervals (CIs); (**C**) scatter plot of the effects of uridine-associated SNPs on ALD risk evaluated by five MR methods (IVW, MR-Egger, simple mode, weighted median, and weighted mode), with the *x*-axis and *y*-axis representing SNP effect sizes (β) on uridine levels and ALD risk, respectively; The arrow in the figure represents the visual depiction of the OR value, with the base of the arrow being the minimum value of the 95% Cl and the tip of the arrow being the maximum value of the 95% Cl.

**Figure 4 ijms-26-05473-f004:**
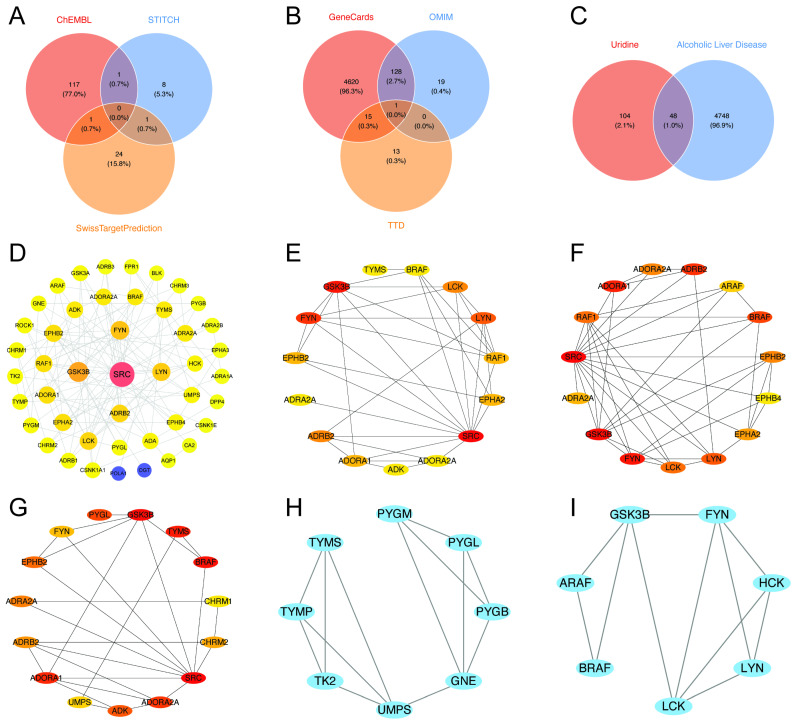
Integrated multi-database analysis of the molecular mechanisms linking uridine and ALD. (**A**) Venn diagram shows the target genes of uridine predicted by ChEMBL, STITCH, and SwissTargetPrediction databases, taking and summing the predictions; (**B**) Venn diagram illustrates the target genes of ALD predicted by GeneCards, OMIM, and TTD databases, taking and summing the predictions; (**C**) Venn diagram illustrates the overlapping target genes of uridine and ALD; (**D**) global topology of the protein–protein interaction (PPI) network; (**E**–**G**) node centrality analysis: key proteins were identified based on node degree (**E**), closeness centrality (**F**), and betweenness centrality (**G**), with SRC ranking within the top 5% across all three metrics, indicating its hub status; (**H**,**I**) using the MCODE plugin in Cytoscape to score 48 core genes, with Module 1 (score = 12.8, (**H**)) and Module 2 (score = 9.6, (**I**)).

**Figure 5 ijms-26-05473-f005:**
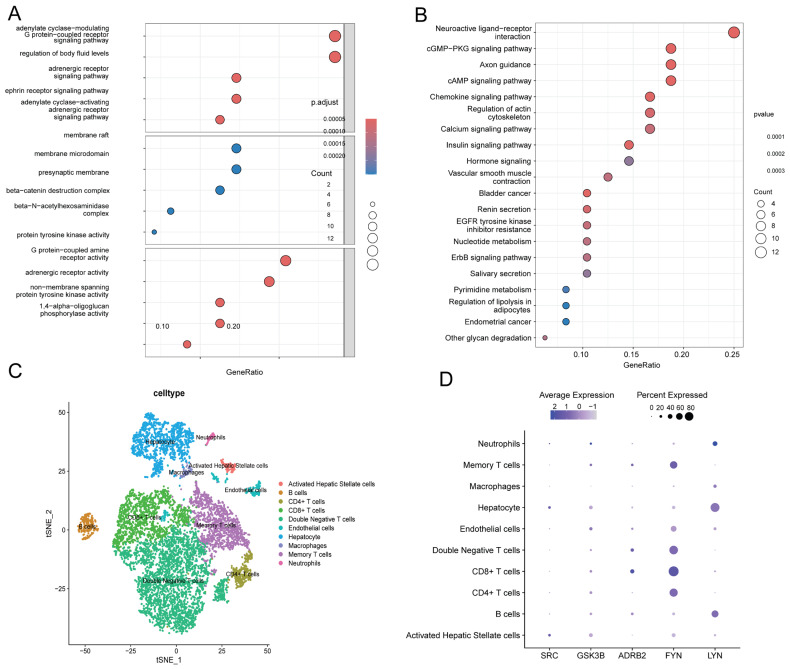
The expression characteristics of 48 core genes were determined by GO enrichment analysis and KEGG enrichment analysis, and the expression characteristics of SRC, LYN, FYN, GSK3B, and ADRB2 were determined by single-cell transcriptome sequencing analysis. (**A**) GO enrichment analysis demonstrates significant enrichment in biological processes and molecular functions such as G protein-coupled receptor (GPCR) signaling regulation and kinase activity. (**B**) KEGG enrichment analysis reveals that the hub genes are primarily associated with nucleotide metabolism, the cGMP-PKG signaling pathway, and cancer-related pathways. The color gradient represents the significance level of the *p*-value. (**C**) Based on the t-SNE algorithm, the dimensionality reduction analysis of cell types was carried out, and the horizontal axis (tSNE_1) and vertical axis (tSNE_2) showed the distribution of different cell types. (**D**) SRC, GSK3B, ADRB2, FYN, and LYN gene expression characteristics: purple dots represent the average expression level (color intensity) and expression percentage (dot size) of genes in various cell types, while dark large dots indicate high expression levels and widespread expression. Note: GSK3B is a gene name, and the protein product is GSK3β.

**Figure 6 ijms-26-05473-f006:**
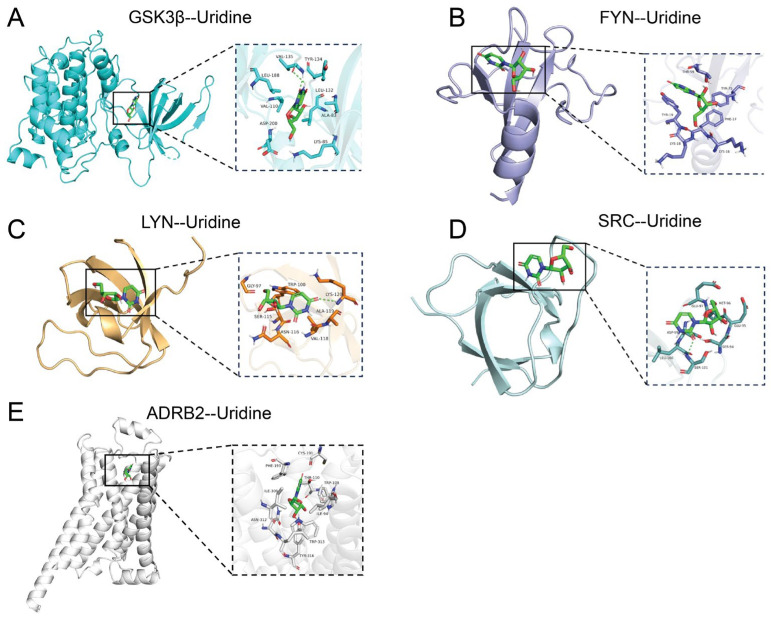
Molecular docking analysis of uridine with key targets. (**A**) GSK3β and uridine molecular docking macro and micro diagrams; (**B**) FYN and uridine molecular docking macro and micro diagrams; (**C**) LYN and uridine molecular docking macro and micro diagrams; (**D**) SRC and uridine molecular docking macro and micro diagrams; (**E**) ADRB2 and uridine molecular docking macro and micro diagrams.

**Figure 7 ijms-26-05473-f007:**
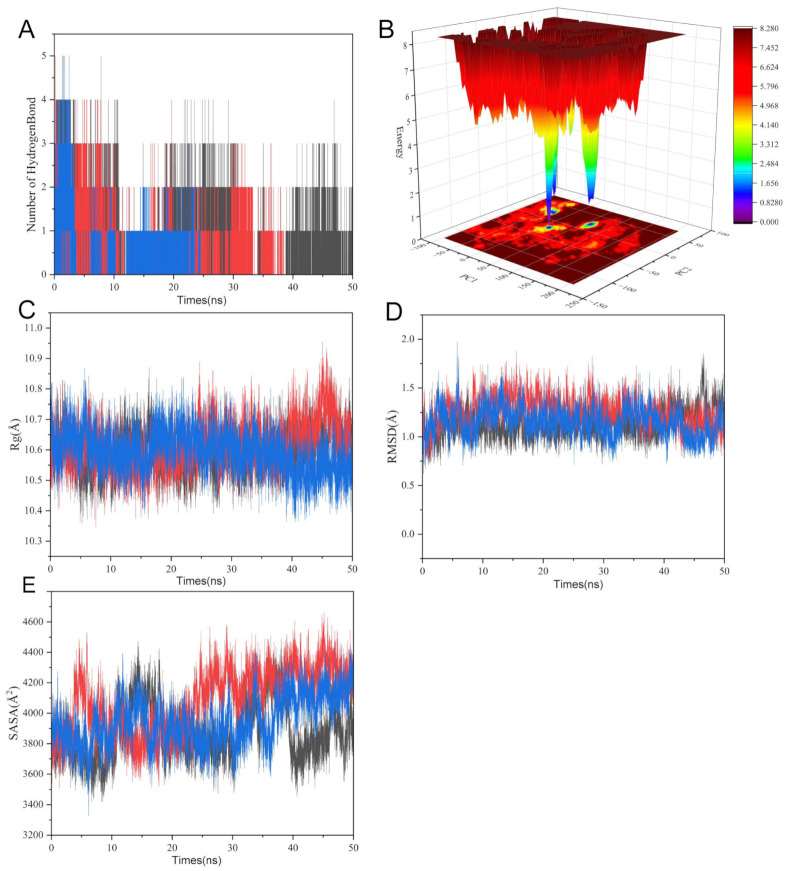
Molecular dynamics analysis of uridine–SRC kinase binding stability. (**A**) Hydrogen bond occupancy highlights GLU-95 stabilizing uridine’s ribose oxygen. (**B**) Free energy landscape along PC1/PC2 dimensions reveals dominant low-energy conformational state. (**C**) Rg (10.6 ± 0.2 Å) stabilization indicates kinase domain compaction. (**D**) RMSD convergence (1.5 ± 0.3 Å post 20 ns) confirms structural equilibration. (**E**) SASA (4000 ± 400 Å^2^) stabilization indicates kinase domain compaction. The red and blue in the image represent SRC and uridine.

## Data Availability

Outcome data from Mendelian randomization from https://gwas.mrcieu.ac.uk/datasets/finn-b-ALCOLIVER/ and exposure data from Supplementary Tables in https://pmc.ncbi.nlm.nih.gov/articles/PMC7614162/ “EMS159509-supplement-Supplementary_Tables.xlsx” (accessed on 13 December 2024). We used the https://tox.charite.de/protox3 (accessed on 11 January 2025) and http://www.swissadme.ch (accessed on 15 January 2025) websites to predict the toxicity of uridine. Potential targets of uridine were predicted using the https://www.ebi.ac.uk/chembl/ (accessed on 23 February 2025), http://www.swisstargetprediction.ch/ (accessed on 25 February 2025), and http://stitch.embl.de/cgi/network.pl websites (accessed on 18 March 2025). Potential targets for ALD were predicted using https://www.genecards.org/ (accessed on 22 March 2025), https://omim.org/ (accessed on 22 March 2025), and https://www.disgenet.org/ (accessed on 22 March 2025), and a GSE236382 single-cell dataset for performing liver disease was obtained from the GEO database.

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
