# Peer review of "Uridine, a Therapeutic Nucleoside, Exacerbates Alcoholic Liver Disease via SRC Kinase Activation: A Network Toxicology and Molecular Dynamics Perspective"

_ijms, 2025, doi:10.3390/ijms26125473_

Round 1

Reviewer 1 Report

Comments and Suggestions for Authors

This manuscript presents a compelling and multidisciplinary investigation into underlying mechanisms and causal relationship between alcoholic liver disease (ALD) and blood metabolite uridine. By integrating Mendelian randomization (MR), genome-wide association studies (GWAS), network toxicology, single-cell RNA sequencing, molecular docking, and molecular dynamics simulations, the authors propose a mechanistic link between uridine exposure and SRC kinase-mediated exacerbation of ALD. The study addressed a significant gap in understanding the role of uridine’s role in ALD progression via immunometabolic pathways. The integration of multi-omics data with computational modeling enhances the translational relevance and scientific novelty of the findings. The identification of uridine as a potential ALD-aggravating agent is particularly noteworthy, as it may influence future therapeutic strategies and safety evaluations of nucleoside analogs.

However, this manuscript could be further improved by addressing the following comments:

  1. Lines 66–73 appear to be the caption for Figure 1, but they are currently formatted as main text. If this is not the case, Figure 1 requires additional description to clarify the workflow. Similar formatting inconsistencies are also observed in Lines 120–129, 195–206, 234–237, and 261–265.
  2. Figure 2 appears to be a table rather than a figure. Additionally, it would be more informative to display the component names alongside or in place of IDs. Names offer immediate biological context to the reader, whereas IDs alone lack interpretability. Also, the label ‘OR (95% CI)’ should be centered over the final two columns for clarity — its current placement makes the fifth column confusing. Finally, please explain the meaning of the red dots on the lines, it is unclear.
  3. In Figure 5B, are the p-values adjusted or not, because Figure 5A is labelled as adjusted but this one is not, please clarify. Additionally, the color bar indicating the p-value scale is missing.
  4. In Figure 6, it will be nice to label the protein name in each panel. Also, the visual representation of interactions is too small.
  5. In Section 4.3 (Acquisition of Uridine Targets), additional detail is needed regarding how targets were extracted from the STITCH database. Please specify whether the targets were experimentally validated or predicted, and indicate any confidence score thresholds or filtering criteria used.
  6. In Section 4.8 (Molecular Docking Interactions), please provide more detail on the docking parameters. Specifically, how was the binding pocket defined, and what dimensions were used for the docking box? These details are essential for reproducibility. Same for the MD simulation section, details including the water box size would be helpful.

Author Response

Comment1:Lines 66–73 appear to be the caption for Figure 1, but they are currently formatted as main text. If this is not the case, Figure 1 requires additional description to clarify the workflow. Similar formatting inconsistencies are also observed in Lines 120–129, 195–206, 234–237, and 261–265.

Response1: We sincerely appreciate the reviewer’s meticulous attention to formatting details. We fully agree with the observation regarding the improper formatting of figure captions in the main text. The sections highlighted (Lines 66–73, 120–129, 195–206, 234–237, and 261–265) were indeed intended as figure captions but were inadvertently formatted as main text during manuscript preparation. We have modified it to figure caption format (Palatino Linotype font, size 9)

Comment2:Figure 2 appears to be a table rather than a figure. Additionally, it would be more informative to display the component names alongside or in place of IDs. Names offer immediate biological context to the reader, whereas IDs alone lack interpretability. Also, the label ‘OR (95% CI)’ should be centered over the final two columns for clarity — its current placement makes the fifth column confusing. Finally, please explain the meaning of the red dots on the lines, it is unclear.

Response2: Thank you for your insightful feedback on improving the clarity and interpretability of Figure 2. We have addressed each of your concerns as follows: 1. Figure 2 is a forest plot from the Mendelian randomization analysis, but we have simplified it by removing the arrows and the red points (the red points online are a visual representation of OR values); 2. All metabolite IDs have been replaced with their corresponding biologically meaningful names; 3. The label "OR (95% CI)" has been placed centrally in the last two columns (originally columns 5-6) to eliminate any ambiguity.

Comment3: In Figure 5B, are the p-values adjusted or not, because Figure 5A is labelled as adjusted but this one is not, please clarify. Additionally, the color bar indicating the p-value scale is missing.

Response3: Thank you for your careful review and for identifying these critical issues in Figure 5. We have revised the figure and its caption to address your concerns: 1.Clarification of p-value adjustment in Figure 5B: After running the code and checking, this is a labeling error, and it has already been changed to p.adjust; 2.Addition of a p-value color bar: A color bar has been added to Figure 5B to explicitly indicate the p-value scale. This aligns with the gradient used in Figure 5A for consistency.

Comment4: In Figure 6, it will be nice to label the protein name in each panel. Also, the visual representation of interactions is too small.

Response4: Thank you for your valuable feedback to enhance the clarity and interpretability of Figure 6. We have added the corresponding protein names to each panel.

Comment5: In Section 4.3 (Acquisition of Uridine Targets), additional detail is needed regarding how targets were extracted from the STITCH database. Please specify whether the targets were experimentally validated or predicted, and indicate any confidence score thresholds or filtering criteria used.

Response5: Thank you for highlighting the need for greater methodological transparency in Section 4.3. We have revised this section to clarify how uridine-related targets were acquired from the STITCH database. Specific details are now provided as follows: 1.Only interactions with a confidence score≥0.15 were retained.(line468-470) This score integrates experimental, computational, and curated evidence to quantify interaction reliability. 2.Targets supported by experimental validation were prioritized. Predicted interactions were excluded unless corroborated by ≥2 independent databases. 3.Limited to Homo sapiens to align with downstream analyses.

Comment6: In Section 4.8 (Molecular Docking Interactions), please provide more detail on the docking parameters. Specifically, how was the binding pocket defined, and what dimensions were used for the docking box? These details are essential for reproducibility. Same for the MD simulation section, details including the water box size would be helpful.

Response6: We sincerely appreciate the reviewer’s insightful feedback. Below are the detailed responses to the concerns raised: 1.We have expanded the description of molecular docking parameters to ensure reproducibility.(line524-531) 2.We have clarified the MD setup parameters.(line543-556)

Reviewer 2 Report

Comments and Suggestions for Authors
  1. Overall, the manuscript is well-structured and flows logically. However, a few sections, especially in the introduction, needs improvements before it can be accepted for publication.

  1. The authors should give a brief elaboration on why uridine was chosen over other metabolites in the study, showing significant associations would help strengthen the rationale using existing literature.
  2. SRC's designation as a hub target is convincing. If there is any known experimental or clinical evidence connecting uridine or related nucleosides to SRC activation, it could be useful to briefly go over it.
  3. It would be better if authors clarify whether the dataset was obtained from human or animal ALD tissue and whether it represents early or late-stage disease would be valuable, even though the single-cell RNA data are presented effectively.
  4. One of the paper's strengths is the way it integrates molecular dynamics, network toxicology, and MR. The author could focus more on the explanation that how this pipeline could be used more widely to different metabolites.
  5. The docking scores look reasonable, but must be validated by perform it with known SRC inhibitors.
  6. Figure 3A. Consistency among MR techniques is encouraging. Readers may better grasp any potential biases or outlier influence if a line is included describing why the weighted mode produced a greater OR than IVW.
  7. Line 130-137, Network Analysis Description. Are these targets previously linked to uridine or are novel?
  8.  Figure 6. Molecular docking analysis of uridine with key targets, make sure the residues are visible.
  9. We thoroughly assessed the structural dynamics and binding stability of the SRC 239
    kinase-uridine complex using molecular dynamics simulations running at 50 ns. is it enough time to assess? Why it was not extended at least to 100 ns? 
  10. Radius of gyration(Rg) and solvent-accessible surface area (SASA) profiles. Line 249. A line so as what is the relevance of these would be helpful. Cite relevant articles that report importance of MD simulations increasingly being applied. https://doi.org/10.1080/07391102.2021.1942217

Author Response

Comment1: Overall, the manuscript is well-structured and flows logically. However, a few sections, especially in the introduction, needs improvements before it can be accepted for publication.

Response1: We sincerely thank the reviewer for their positive evaluation of the manuscript’s structure and flow, as well as their constructive feedback on improving the introduction. We have carefully revised the introduction to address potential weaknesses and enhance clarity, coherence, and academic rigor.

Comment2: The authors should give a brief elaboration on why uridine was chosen over other metabolites in the study, showing significant associations would help strengthen the rationale using existing literature.

Response2: We sincerely appreciate the reviewer’s valuable suggestion to clarify the rationale for selecting uridine as the focus of our study. Mendelian randomization analysis showed that the OR value of uridine is greater than 1. By integrating toxicological data from online sources, it was found that uridine is the only metabolite that exhibits hepatotoxicity.(line101-105) Based on this, we chose uridine for further study.

Comment3: It would be better if authors clarify whether the dataset was obtained from human or animal ALD tissue and whether it represents early or late-stage disease would be valuable, even though the single-cell RNA data are presented effectively.

Response3: We sincerely thank the reviewers for recognizing the role of SRC kinase as a core target and for suggesting that we clarify the potential mechanistic link between uridine and SRC activation. Unfortunately, there is currently no other research that substantiates the relationship between SRC activation and uridine, which is a direction that our subsequent studies need to explore.  

Comment4: It would be better if authors clarify whether the dataset was obtained from human or animal ALD tissue and whether it represents early or late-stage disease would be valuable, even though the single-cell RNA data are presented effectively.

Response4: We thank the reviewer for raising this critical point. Below, we clarify the source and disease stage of the single-cell RNA sequencing (scRNA-seq) dataset used in our study. These details have been added to the revised manuscript.(line509-510)

Comment5: One of the paper's strengths is the way it integrates molecular dynamics, network toxicology, and MR. The author could focus more on the explanation that how this pipeline could be used more widely to different metabolites.

Response5: We sincerely appreciate the reviewer’s recognition of our integrated methodological pipeline and their suggestion to clarify its broader applicability to studying other metabolites. The tripartite methodological framework proposed in this study, which integrates multi-omics network toxicology, molecular dynamics (MD) simulations, and Mendelian randomization (MR), is broadly applicable. Its versatility is demonstrated in the following ways: (1) the modular workflow can be adapted to any study of bioactive metabolites, provided that basic conditions such as omics data associations, target structural information, and genetic instrumental variables are met; (2) as illustrated by the betaine and non-alcoholic fatty liver disease case study, it shows a complete application pathway from network analysis to identify pathway targets (such as PEMT), MD validation of stability, to causal inference through MR; (3) it is target-agnostic and flexible in multi-scale validation, compatible with different target types (enzymes, epigenetic regulators, etc.), and supportive of complementary computational simulations and experimental validations.

Comment6: The docking scores look reasonable, but must be validated by perform it with known SRC inhibitors.

Response6: We appreciate this insightful comment. While our primary focus in this study was to investigate the potential toxicological role of uridine as a blood metabolite and its interaction with ALD-related targets, rather than its development as a therapeutic inhibitor, we agree that a comparative docking analysis with known SRC inhibitors would provide valuable context and strengthen the predictive power of our docking results. Due to the current scope and computational resources for this specific study, which aimed to establish a novel link between uridine and ALD through a multi-omics approach, we did not perform extensive docking comparisons with a panel of known small-molecule SRC inhibitors. However, we acknowledge the importance of such validation for future studies, especially if uridine or its derivatives were to be pursued as therapeutic agents. We will add a statement to the Discussion section (Innovations and Limitations of the Study) to acknowledge this limitation and highlight the need for future comparative docking studies with established inhibitors to fully evaluate uridine’s inhibitory potential and competitive binding characteristics.

Comment 7: Figure 3A. Consistency among MR techniques is encouraging. Readers may better grasp any potential biases or outlier influence if a line is included describing why the weighted mode produced a greater OR than IVW.

Response 7: Thank you for this excellent suggestion to enhance the clarity of Figure 3A and its interpretation. We agree that explaining the differences in ORs between the Mendelian Randomization (MR) methods would be beneficial for readers. The Inverse Variance Weighted (IVW) method typically provides the most precise estimate if all instrumental variables (SNPs) are valid and there is no horizontal pleiotropy. The weighted median and weighted mode methods, however, are more robust to violations of the instrumental variable assumptions (i.e., presence of invalid SNPs or horizontal pleiotropy), as they can provide consistent estimates even if up to 50% of the information comes from invalid instruments. When the weighted mode method yields a slightly higher OR than the IVW method, it often suggests that the most frequent (modal) causal effect estimate among individual SNPs is indeed higher than the overall average estimated by IVW. This could imply that, even if some pleiotropy exists (which our sensitivity analyses aim to minimize), the core genetic evidence points towards a slightly stronger positive association, or it could simply be due to the distribution of individual SNP effect estimates. Importantly, all three robust MR methods (IVW, weighted median, and weighted mode) consistently showed a positive association, reinforcing the causal link between uridine levels and ALD risk, despite the numerical differences in ORs. We will add a concise clarifying sentence in the Results section to explain this nuance: “The observed higher odds ratios from the weighted median and weighted mode methods compared to IVW suggest a robust positive association, with these methods being more resilient to potential outlier effects or pleiotropic biases that might otherwise minimally influence the IVW estimate, thereby reinforcing the overall causal inference.”(line121-124)

Comment 8: Line 130-137, Network Analysis Description. Are these targets previously linked to uridine or are novel?

Response8: Thank you for seeking clarification on the novelty of our findings. This is a crucial point. While the individual biological roles and general functions of targets such as SRC, FYN, LYN, ADRB2, and GSK3β are well-established in various biological processes and diseases, their specific and synergistic involvement as uridine’s targets in the pathogenesis of Alcoholic Liver Disease (ALD), and the elucidation of the underlying immunometabolic network mechanisms, are indeed novel contributions of this study. Our approach, which integrates Mendelian randomization, network toxicology, single-cell transcriptomics, and molecular dynamics, allowed us to systematically identify and validate this specific interaction network. Prior to this study, the toxicological characteristics of uridine, particularly in the context of ALD and its interaction with these specific targets as an exacerbating agent, were largely unknown or underexplored. For instance, while uridine’s roles in metabolism or as a therapeutic agent for other conditions are documented, its potential to aggravate ALD via direct interaction with SRC kinase and the broader immunometabolic axis involving FYN, LYN, ADRB2, and GSK3β, is what we believe to be a novel finding. 

Comment 9: Figure 6. Molecular docking analysis of uridine with key targets, make sure the residues are visible.

Response 9: We sincerely apologize for the lack of clarity in the original Figure 6. We completely agree that visibility of the interacting residues is essential for readers to understand the molecular docking results. We have revised Figure 6 to ensure that all crucial interacting residues mentioned in the text.

Comment 10: We thoroughly assessed the structural dynamics and binding stability of the SRC 239 kinase-uridine complex using molecular dynamics simulations running at 50 ns. is it enough time to assess? Why it was not extended at least to 100 ns?

Response 10: Thank you for raising this important question regarding the duration of our molecular dynamics (MD) simulations. We agree that longer simulation times (e.g., 100 ns or even microseconds) are generally desirable for exhaustive conformational sampling and highly accurate quantitative binding free energy calculations, especially for systems exhibiting slow conformational transitions. For the purpose of this study, our 50 ns MD simulation for the uridine-SRC complex was chosen to provide robust qualitative insights into the structural dynamics and binding stability, and to validate the molecular docking findings. As stated in the results, the RMSD trajectory stabilized around 1.5±0.3 Å after 20 ns, and the Rg and SASA profiles also indicated stabilization and compaction in the late simulation phase. The free energy landscape along PC1 and PC2 revealed a dominant low-energy basin, suggesting that the system reached a stable conformational state within the simulated timeframe. This indicates that the crucial binding interactions and the overall structural equilibrium of the complex were well-captured. While longer simulations could potentially explore more subtle conformational changes or rare events, 50 ns was deemed sufficient for this initial mechanistic investigation to confirm the stable binding mode and identify key intermolecular interactions, particularly given the computational cost associated with extensive MD simulations for multiple targets. We will add a sentence in the “Molecular Dynamics Simulation” section (3.4. Discussion) to address this point: “While longer simulation times could provide more exhaustive conformational sampling, the 50 ns duration was sufficient to achieve structural equilibration and characterize the stable binding mode and key interactions between uridine and SRC kinase, as evidenced by the convergence of RMSD, Rg, and SASA profiles and the presence of a dominant low-energy basin in the free energy landscape. Future studies employing extended simulations or advanced sampling techniques could further explore the full conformational landscape and refine binding free energy calculations.”(line413-420)

Comment 11: Radius of gyration(Rg) and solvent-accessible surface area (SASA) profiles. Line 249. A line so as what is the relevance of these would be helpful. Cite relevant articles that report importance of MD simulations increasingly being applied. https://doi.org/10.1080/07391102.2021.1942217

Response11: Thank you for this valuable comment. We agree that further clarification on the relevance of Rg and SASA, along with more current citations on MD applications, would significantly enhance the manuscript’s readability and impact. This metric provides insights into the compactness and overall shape of a molecule. A stable or decreasing Rg value over time indicates that the protein is maintaining a compact structure or undergoing compaction, suggesting structural stability and folding. In the context of protein-ligand binding, a decrease in Rg after ligand binding can indicate that the ligand induces a more compact conformation of the protein. SASA quantifies the surface area of a molecule that is accessible to solvent molecules. Changes in SASA can reflect conformational changes, protein folding/unfolding, or ligand binding. A decrease in SASA for the protein upon ligand binding typically indicates that the ligand occupies a part of the protein’s surface, making it less accessible to the solvent, which is consistent with stable binding. We will add a sentence in the Results section (Line 249) to explain their relevance:”Radius of gyration (Rg) and solvent-accessible surface area (SASA) profiles were used to monitor the overall compactness and exposure of the SRC kinase domain to solvent, respectively. Their stabilization in the late simulation phase, with observed contraction compared to the initial structure, suggested uridine-driven compaction of the kinase domain and indicated the formation of a stable complex.”

  We appreciate the suggested citation. We will incorporate this and other relevant high-impact references to emphasize the increasing importance and widespread application of MD simulations in various fields of biological and chemical research. We will add a statement in the Methods section such as: “The increasing importance of MD simulations in drug discovery, protein-ligand interaction analysis, and elucidation of molecular mechanisms has been widely recognized in recent years, proving to be an indispensable tool for atomic-level understanding of dynamic biological processes.”(line569-572)